# Pain in Multiple Sites and Clusters of Cause-Specific Work Disability Development among Midlife Municipal Employees

**DOI:** 10.3390/ijerph18073375

**Published:** 2021-03-24

**Authors:** Aapo Hiilamo, Anna Huttu, Simon Øverland, Olli Pietiläinen, Ossi Rahkonen, Tea Lallukka

**Affiliations:** 1Finnish Institute of Occupational Health, PO Box 18, 00032 Helsinki, Finland; tea.lallukka@helsinki.fi; 2Department of Public Health, University of Helsinki, PO Box 20, FI-00014 Helsinki, Finland; anna.huttu@icloud.com (A.H.); olli.k.pietilainen@helsinki.fi (O.P.); ossi.rahkonen@helsinki.fi (O.R.); 3Division of Physical and Mental Health, Norwegian Institute of Public Health, N-0403 Oslo, Norway; SimonNygaard.Overland@fhi.no

**Keywords:** work disability, pain, public sector

## Abstract

This study investigates to what extent pain in multiple sites and common risk factors related to work environment, occupational class and health behaviours are associated with cause-specific work disability (WD) development clusters. The study population was derived from the Finnish Helsinki Health Study (n = 2878). Sequence analysis created clusters of similar subsequent cause-specific WD development in an eight-year follow-up period. Cross-tabulations and multinomial logistic regression were used to analyze the extent to which baseline factors, including pain in multiple sites, were associated with the subsequent WD clusters. A solution with five distinct WD clusters was chosen: absence of any WD (40%), low and temporary WD due to various causes (46%), WD due to mental disorders (3%), WD due to musculoskeletal (8%) and WD due to other causes (4%). Half of the employees in the musculoskeletal WD cluster had pain in multiple locations. In the adjusted model the number of pain sites, low occupational class and physical working conditions were linked to the musculoskeletal WD. The identified characteristics of the different WD clusters may help target tailored work disability prevention measures for those at risk.

## 1. Introduction

Pain emerges as one of the most important early predictors of a long-term work disability due to musculoskeletal disorders and also other causes [1,2,3]. This calls for more detailed investigations on when and what causes pain that leads to work disability. The reduction of work disability (sickness absence (SA) and disability pension (DP)) is a widely acknowledged policy goal in Finland, one of the countries in which the number of working aged people is decreasing [4]. Thorough understanding of the predictors of work disability is vital for this policy goal.

Previous studies, mainly analysing either the number of sickness absences (SA) or time to first long-term work disability event (DP) have shown that the locations, number of areas and intensity of pain contribute to the risk of subsequent long-term work disability [1,2,5,6,7,8]. A Finnish nationally representative study found that a number of musculoskeletal pain sites predict a higher risk of a disability pension award in a dose-response fashion [1]. Similar findings are reported in Norwegian population-based studies where widespread pain [2] and number of pain sites [8] predicted higher risk of work disability. However, the previous studies on pain and work disability have rarely analysed sickness absence and disability pensions simultaneously, taken the heterogeneity in the work disability development into account or analysed specific pain areas. 

Given the availability of longitudinal and retrospective data, sequence analysis has been increasingly used to provide a more comprehensive picture of individual trajectories in successive states [9]. Sequence analysis techniques are often used, for example, to analyse employment patterns after a vocational rehabilitation [10,11,12]. However, the use of sequence analysis to investigate cause-specific work disability development is still limited. The heterogeneity in the diagnostic causes, interrelations, transitions, timing and the persistence of work disability is challenging to summarize while using traditional expected number of spells or time-to-event type modelling approaches. In contrast, sequence analysis can summarize this heterogeneity by grouping individuals with a similar cause-specific work disability development over time [9]. The method can take the duration of different states into consideration [9], which implies that a data-driven distinction between temporary and permanent work disability can be made. Simplifying cause-specific work disability development to meaningful clusters and identifying their determinants can provide added value for potential tailored prevention strategies. 

The aim of the present study is to determine the associations between pain in multiple sites and clusters of cause-specific work disability development among midlife employees. To address the aim, we first study clusters of cause-specific work disability development over an eight-year follow-up period among initially full-time midlife employees. Second, we investigated to what extent baseline pain in multiple sites and other covariates are associated with the membership to the cause-specific work disability clusters. 

The longitudinal data on pain and work disability were analysed in this cohort previously [13]. The present analysis expands on this work in several ways: first, we combine sickness absence and disability pension to single variable; second, the main exposure variables of interest in the present study are specific pain areas and multisite pain, as measured in 2007; third, we now use a sequence analysis approach for the work disability outcome. Sequence analysis enables us to summarize the heterogeneity in work disability development while taking the length and interrelation of cause-specific work disabilities into consideration. 

## 2. Materials and Methods

### 2.1. Data

The sample included for the analyses consisted of 2878 initially full-time employees aged 45–57 in 2007, derived from the Helsinki Health Study (HHS). This cohort study is described in detail elsewhere [14]. In brief, HHS focusses on midlife employees of the City of Helsinki, the largest employer and municipality in Finland. The original baseline of the study was in 2000-2002 when mailed questionnaires were sent to all employees of the City of Helsinki reaching their 40, 45, 50, 55 and 60 birthdays in each year. The cohort includes the employer’s and national register linkages for the respondents who provided consents for such linkages. This analysis uses a follow-up survey conducted in 2007 as the baseline for this study, given that data on cause-specific sickness absence were not available immediately after the original baseline survey in 2000-2002.

The inclusion criteria for the present study were the following: participants who provided informed consents to register linkages in phase 1 survey in 2000–2002 and who were respondents in the subsequent survey conducted in 2007 (response rate 83%; this was used as a baseline survey for the current study to address the aims of the study). It was also required that persons were full-time employees at the time of the survey, had no previous disability pension award, had less than 180 days of work disability in the year before the survey and had no old-age or early pension award in the first five years since the survey (those with an earlier old-age pension award were not included given that their actual risk for work disability was substantially shorter). See
Appendix A for more details in sample selection.

### 2.2. Ethics

Department of Public Health, University of Helsinki and the City of Helsinki, Finland, gave ethical approval for the Helsinki Health Study.

### 2.3. Measures: Diagnostic Specific Work Disability

Data on work disability were obtained from two national social insurance registers. The Social Insurance Institution of Finland (Kela) provided cause-specific information on sickness absence periods lasting more than 10 working days. The Finnish Centre for Pensions (ETK) provided cause-specific information on all pension awards. These social insurance administrative sources can be regarded as highly reliable as they require medical certificates for the disability awards. Three causes of work disability were distinguished. Musculoskeletal-related work disability was defined as a work disability due to ICD-10 codes M00–M99 (first diagnostic reason in work disability pension awards), mental due to F00-F99 and other due to other causes. The work disability follow-up lasted eight years after the survey (see the Statistical analysis section).

### 2.4. Predictors: Pain in Multiple Sites

All predictors were obtained from the postal survey conducted in 2007, before the work disability follow-up. We used the number of reported pain sites as a marker of pain severity because previous evidence shows that it strongly predicts the extent to which pain leads to work disability [6,7,8,15]. The sample size was too small to reliably examine combinations of different sites of pain across the body, although they might capture different aspects and a broader understanding as compared to the number of pain sites. The employees were asked in the questionnaires whether they were currently experiencing pain, and six specified and one unspecified body locations in which the pain was experienced (head/face, neck/shoulders, low back, lower limbs, upper limbs, stomach and a self-reported location). Based on previous research, we divided pain to no pain, single-located pain, pain in two locations and pain in three or more locations. Furthermore, we also provided cross-tabulations on the specific pain locations and the work disability clusters. 

### 2.5. Other Predictors

We included a set of commonly known sociodemographic, health, work- and behaviour-related risk factors of work disability. Sociodemographic factors included age, sex and occupational class. Age was used as a categorical variable (groups 45–49, 50–54 and 55–57) to allow a potentially nonlinear association between age and work disability cluster memberships. Participants’ occupational title was obtained from the employer’s register, or for those without such information from the postal questionnaires. Following conventional procedures, we classified occupations into four groups (managers and professionals, semiprofessionals, routine nonmanuals and manual workers). Multivariable model was also adjusted for gender, as sample size was too small to analyse men and women separately.

Work environment variables included dichotomized perceived physically and mentally strenuous work environment. Behavioural risk-factors included binge drinking (more than once a month), smoking (nonsmoker, previous, current), body weight (healthy weight (BMI < 25, overweight BMI 25–30, obese BMI ≥ 30). Finally, mental health was measured by General health questionnaire GHQ [16] with a cut-point of 3 or more, following previous procedures [17]. 

### 2.6. Statistical Analysis

Cause-specific work disability development was investigated using a sequence analysis approach. The time-unit for work disability measurement was in single years since submitting the 2007 survey (i.e., t0 = from the date submitting the survey to +365 days from this date, t1 = from the t0 endpoint to +365 days, and so forth). Our follow-up time lasted eight years, yielding eight observations for each individual (positions t0…t7). In each position, work disability states consisted of: (a) no work disability, (b) work disability due to mental causes, (c) due to musculoskeletal causes and (d) due to other causes. These states were defined as having at least one corresponding work disability event (long-term sickness absence or being on a disability retirement) during a given time-unit. The four states were mutually exclusive. If a person had work disability events due to more than one cause, the work disability state was determined by the cause due to which more working days were lost within that time-unit (year). For those who deceased before the time-unit (32 persons), work disability state was imputed as the work disability state the year before the death. 

To identify clusters of a similar development in cause-specific work disability, clustering techniques were used. The first cluster was manually formed, which consisted of the employees without any work disability during the follow-up period. Other employees with at least one work disability period were included in the cluster analysis. We first computed similarities between sequences using the longest common subsequence (LCS) measure [18,19]. This measure emphasized the length and order of the states, rather than timing (years since the survey), which was not relevant in the present study. Using the similarity matrix as an input, clusters of cause-specific work disability development were created using Ward’s clustering algorithm [20]. The optimal number of clusters was guided by average silhouette width (as a measure of cluster quality), reasonable cluster sizes and meaningful interpretation. The cluster quality measures and other illustrations supporting the distinct interpretation of each cluster are in the Appendix A.

The composition of the work disability clusters was then examined with cross-tabulations. In cross-tabulations, we used chi^2^ test when appropriate. Furthermore, a multinomial logistic regression model was fitted to examine predictors of the cluster group membership after adjusting for other variables. We conducted a single multivariable model, consisting of all employees. Average marginal effects (AMEs) for each cluster group were reported. In Appendix A we report odds ratios using the no-work disability group as reference group (Appendix A). 

Some 10% of the study population had missingness in at least one of the predictors. Multiple imputations by chained equations with 50 datasets was chosen as the strategy to impute missingness for the multinomial logistic regression models. We used all the covariates and the outcome variable in the imputations process. Sequence analysis was conducted using R and TraMiner package [21] and rest with Stata 15.

## 3. Results

Figure 1 shows the distributions of work disability states by time-unit and by five clusters that were identified in the cluster analysis: 1. *no work disability cluster* (*n* = 1138/40%), 2. Temporary and *minor work disability* due to various reasons (*n* = 1313/46%, hereafter *minor work disability cluster*), 3. Work disability due to *mental* causes (*n* = 78/3%, hereafter *mental work disability cluster*), 4. Work disability due to *musculoskeletal* causes (*n* = 238/8%, hereafter *musculoskeletal work disability cluster*) 5. Work disability due to *other reasons* (*n* = 111/4%, hereafter *other work disability cluster*). 

The characteristics of individuals classified in the five clusters are shown in Table 1. Women were less likely to belong to the no work disability cluster and more likely to belong to the minor work disability cluster and to the musculoskeletal work disability cluster. The clusters differed in the occupational class distribution. For example, 14% of manual workers and of routine nonmanual workers were in the musculoskeletal work disability cluster, whereas this figure for managers and professionals was 3%. However, there were no consistent occupational class differences in the mental cluster.

The five clusters differed in the prevalence of pain and other risk factors (Table 2). While 67% of persons in the no work disability cluster did not report any pain, this figure was 54% in the minor work disability cluster and only 29% in the musculoskeletal work disability cluster. The prevalence of pain in head or face was highest in the mental work disability cluster and other type cluster (13%). In the musculoskeletal work disability cluster, the share of reporting pain in neck or shoulder (41%) or in lower limbs (39%) was the highest. People in the mental work disability cluster reported often a mentally strenuous working environment, whereas the share of reporting physically strenuous working environment was highest in the musculoskeletal work disability cluster. In the other work disability cluster, binge drinking (33% compared to 23% in the no work disability cluster) and smoking (36% vs. 13%) were common.

After mutual adjustments, the number of pain sites was linked to a decreasing likelihood of the no work disability cluster in a dose-response fashion (Table 3). Compared to those without any pain, pain in three or more locations was linked to some 17 percentage points lower predicted probability of the none work disability cluster, most of which were driven due to a higher predicted probability of the musculoskeletal work disability cluster membership (AME = 8.1 (95% confidence interval 4.7–11.6)). Mentally strenuous working conditions predicted the mental work disability cluster (AME 2.3 (95% CI 0.1–4.5)), whereas physically strenuous working conditions predicted the musculoskeletal work disability cluster (AME = 4.8 (2.4–7.2)).

## 4. Discussion

This study focused on the extent to which baseline pain in multiple sites and other characteristics predicted the clusters of work disability development during an eight-year follow-up period among midlife employees. Five distinct work disability clusters were created, each of which had a meaningful interpretation and were different from one another in terms of causes and length of work disability as well as their predictors. However, pain was a key predictor for each of the work disability clusters.

The two largest clusters were a no work disability cluster (40%) and temporary and minor work disability due to various reasons, a minor work disability cluster (46%). These two clusters accounted for 86% of all employees included in this study. The employees in the no work disability cluster were mainly from higher occupational classes, had normal weight and were less likely to report any pain, adverse working conditions and behavioural risk factors. Pain, occupational class, obesity, binge drinking and smoking differentiated those in the no work disability cluster from those in the minor work disability cluster. These characteristics are in line with the findings from previous research focusing on predictors of sickness absence [22,23].

The three other groups were characterized by a more permanent work disability but due to different causes. The third and smallest cluster was work disability due to mental causes. Pain in three or more sites was common in this mental work disability cluster. This is not a surprise given that comorbidity of mental disorders and pain is well-known; studies from Finland [1], Norway [2] and Sweden [3] have shown that pain reporting predicts disability pension due to mental causes. In multivariable analysis adjusting for covariates, including common mental disorders, pain only in two or more sites remained as a significant predictor of this cluster. In line with this finding, a recent Swedish study showed that pain predicts mental work disability [3] and that the association is particularly strong when pain is in multiple sites.

As expected, common mental disorders and adverse mental working conditions were key predictors of the mental work disability cluster. These characteristics differentiated this cluster from the other two permanent work disability clusters (musculoskeletal and other cause). There was no evidence that those in the manual occupational position were more likely to be in this cluster. The absence of clear occupational class gradient in work disability due to mental causes among older employees in Finland was also documented earlier [24].

The fourth cluster was work disability due to musculoskeletal work disability, which consisted of about 8% of individuals. The prevalence of pain was the highest in this cluster, and pain in multiple sites was also a strong predictor in the adjusted model. Pain in neck and shoulder was common in this cluster (41%). Given the differences in methods and follow-up times, it is challenging to compare effect sizes with previous work. However, similarly to previous studies [1,2,5,6,7,15], having two or more pain areas was linked to a higher risk than just one pain area. In line with previous investigations, for example, [24], the musculoskeletal cluster was strongly related to occupational class, even after adjusting for other risk factors, including physically strenuous working conditions. This implies that targeted actions to prevent musculoskeletal disorder-related permanent work disability are important for routine and manual workers, those working in physically strenuous occupations and those reporting pain in multiple sites.

The fifth and last cluster was characterized by other work disability states, a residual category of causes other than musculoskeletal or mental. Pain in unspecific other locations was more prevalent in this cluster. Other key characteristics were manual occupational class, current smoking and binge drinking.

### Methodological Considerations

This study combined registered data on cause-specific work disability with survey information on pain and other key risk factors. Several limitations, however, must be taken into consideration. Our study population consisted of midlife, mainly female, public sector employees in a country with a comprehensive social insurance system. Therefore, generalization of these results is limited to other types of occupational populations or countries. However, in general, the associations between pain and work disability have been consistent across cohorts in studies using variable-oriented methods [1,2,3,5,6,7,15].

Our measure of pain is not without limitations. Pain is here reported as a subjective experience, with cause unknown [25], and not assessed by a health professional. Furthermore, we did not have detailed data on the recurrence of pain and the length of the pain episodes. Lastly, our findings regarding pain and work disability should be considered in the cohort’s context where midlife, mainly female public sector employees dominate. It is worth noting, however, that multisite pain has shown similar associations with work disability in a cohort representative of all Finnish employees [1,15].

Regarding predictors of the identified clusters, self-reporting bias cannot be ruled out. For example, the self-reported working conditions may be affected by other factors, such as mental health, rather than “objective” working conditions. However, working conditions are, nevertheless, in part subjective and related to one’s own assessment and ability. Furthermore, an important limitation was the inability to focus more in detail to different causes of work disability. We constructed the other work disability category as a residual category, which contains a heterogeneous set of causes other than mental or musculoskeletal. The sample size was too small to analyse more specific causes of work disability separately. Furthermore, individuals were allowed to move to (an early) old age retirement in the last three years of the follow-up (and were thereby not in an actual risk of work disability during these years), which was reflected in a small decline in the prevalence work disability in the last year of the follow-up. Lastly, the identified predictors of work disability cluster do not imply causal relationships between these variables and the work disability clusters. Although the multivariable analysis controlled for other observable factors, other unobservable confounding factors were not controlled for.

## 5. Conclusions

These results suggest that pain is an important early determinant of work disability development due to different diagnostic reasons. Work disability development in midlife can be summarized to clusters, each of which differ in terms of work disability cause and length and the baseline risk factors, including pain in multiple sites. Interventions of work disability prevention may find it beneficial to apply tailored prevention strategies and pain management. As highlighted before [26], this means that the varying needs of workers with pain should be identified and the intervention measures designed as relevant to those needs.

## Figures and Tables

**Figure 1 ijerph-18-03375-f001:**
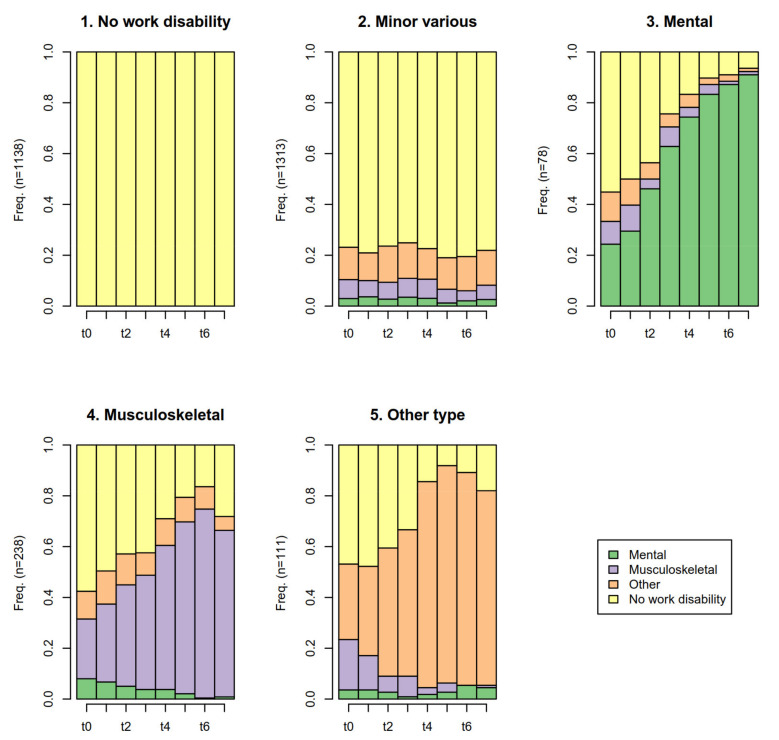
The five identified work disability clusters after submitting the 2007 survey in the Finnish Helsinki Health Study cohort (*n* = 2878). Mean cross-sectional distribution of work disability states in the follow-up. Yellow = no work disability, purple = Musculoskeletal work disability (M00–M99), green = mental work disability (F00–F99), sand = other work disability.

**Table 1 ijerph-18-03375-t001:** Characteristics of the population and the identified work disability clusters.

		Work Disability Clusters				
	N	1. No Work Disability	2. Minor Various	3. Mental	4. Musculoskeletal	5. Other Type
		Row N/%	Row N/%	Row N/%	Row N/%	Row N/%
N	2878	1138	1313	78	238	111
Total	-	40	46	3	8	4
**Gender**						
Men	510	45	45	2	4	4
Women	2368	38	46	3	9	4
**Age group**						
45–49	947	44	45	2	7	3
50–54	1023	37	47	3	8	4
55–57	908	37	45	3	10	4
**Occupational class**						
Managers or professionals	915	51	42	2	3	2
Semi-professionals	693	42	46	4	5	3
Routine non-manual workers	903	31	48	3	14	4
Manual workers	316	26	49	3	14	8
Missing	51	33	53	2	6	6

**Table 2 ijerph-18-03375-t002:** Work disability clusters, pain and modifiable working and lifestyle factors.

Work Disability Clusters						
	1. No Work Disability	2. Minor Various	3. Mental	4. Musculoskeletal	5. Other Type	Chi2
	Col %	Col %	Col %	Col %	Col %	*p*-value
**Number of pain locations**						
No pain	67	54	47	29	36	
Single location pain	15	19	14	21	20	<0.001
2 locations	9	13	6	22	20	
3–7 locations	7	12	31	26	23	
**Pain locations**						
Pain in head or face	4	5	13	9	13	<0.001
Pain in neck or shoulder	17	25	35	41	37	<0.001
Pain in low back	12	17	35	34	25	<0.001
Pain in upper limbs	10	16	19	24	25	<0.001
Pain in lower limbs	11	18	26	39	31	<0.001
Pain in stomach location	3	5	9	5	7	0.018
Pain in some other location	2	4	4	8	12	<0.001
**Mentally strenuous working environment**	11	13	27	13	19	0.003
**Physically strenuous working environment**	23	29	27	55	42	<0.001
**Smoking**						
No	64	56	58	47	41	
Past smoking	22	23	19	22	21	<0.001
Smoking	13	21	23	30	36	
**Binge drinking**						
Binge drinking (once a month or more)	23	28	22	24	33	<0.001
**Obesity**						
Healthy weight	55	47	35	39	34	
Overweight	32	35	37	38	44	<0.001
Obesity	12	17	28	21	21	
**Common mental disorders**						
Common mental disorders	20	24	49	34	39	<0.001
Helsinki Health Study, Finland. Missing categories omitted (1–3%).						

**Table 3 ijerph-18-03375-t003:** Predictors of the work disability clusters. Average marginal effects (AME) and their 95% confidence intervals from multinomial logistic regression. N = 2878. Missing values are imputed.

	Work Disability Clusters				
	1. No Work Disability	2. Minor Various	3. Mental	4. Musculoskeletal	5. Other Type
	AMEs [95% CI]	AMEs [95% CI]	AMEs [95% CI]	AMEs [95% CI]	AMEs [95% CI]
**Pain (ref. no pain)**					
Single location pain	−9.9 *** [−14.6–−5.1]	4.2 [−0.9–9.2]	−0.6 [−2.1–0.8]	4.9 *** [2.1–7.7]	1.5 [−0.5–3.4]
2 locations	−13.7 *** [−19.1–−8.3]	4.5 [−1.2–10.3]	−1.5 * [−2.9–−0.1]	8.1 *** [4.6–11.5]	2.6 * [0.2–5.0]
3–7 locations	−17.1 *** [−22.7–−11.5]	3.7 [−2.3–9.8]	2.4 * [0.0–4.7]	8.1 *** [4.7–11.6]	2.9 * [0.4–5.3]
**Common mental disorders (Ref. no)**					
Yes	−4.5 * [−8.7–−0.3]	−2.0 [−6.4–2.5]	2.7 ** [1.0–4.5]	1.9 [−0.5–4.3]	1.8 * [0.0–3.6]
**Gender (ref. men)**					
Women	−4.7 ^+^ [−9.6–0.3]	2.5 [−2.7–7.6]	0.3 [−1.3–2.0]	2.6 ^+^ [−0.1–5.4]	−0.7 [−2.9–1.4]
**Age in 2007 (ref. 45–49)**					
50–54	−5.1 * [−9.3–−0.9]	1.3 [−3.1–5.7]	1.5 * [0.2–2.9]	0.9 [−1.4–3.2]	1.4 ^+^ [−0.2–3.0]
55–57	−4.8 * [−9.1–−0.5]	−0.4 [−5.0–4.1]	1.3 ^+^ [−0.1–2.7]	2.4 ^+^ [−0.0–4.9]	1.5 ^+^ [−0.2–3.2]
**Occupational class (ref. managers or professional)**					
Semi-professionals	−7.6 ** [−12.5–−2.7]	2.9 [−2.1–7.9]	2.1 * [0.4–3.8]	1.3 [−1.1–3.7]	1.4 [−0.5–3.2]
Routine non-manual workers	−14.6 *** [−19.5–−9.7]	5.7 * [0.6–10.8]	0.6 [−0.8–2.1]	6.8 *** [4.1–9.4]	1.4 [−0.4–3.2]
Manual workers	−19.0 *** [−25.7–−12.3]	8.4 * [1.2–15.5]	1.1 [−1.3–3.4]	5.7 ** [2.0–9.4]	3.8 * [0.8–6.9]
**Working conditions**					
Physically strenuous work (ref. no)	−2.0 [−6.3–2.4]	−2.7 [−7.2–1.9]	−0.9 [−2.3–0.4]	4.8 *** [2.4–7.2]	0.8 [−1.0–2.5]
Mentally strenuous work (ref. no)	−4.5 ^+^ [−9.8–0.8]	1.4 [−4.3–7.0]	2.3 * [0.1–4.5]	−0.6 [−3.5–2.3]	1.4 [−0.9–3.8]
**Body weight (ref. BMI < 25)**					
Overweight	−5.8 ** [−9.6–−1.9]	1.3 [−2.8–5.4]	0.8 [−0.4–2.1]	1.8 [−0.4–4.0]	1.8 * [0.2–3.4]
Obesity	−9.0 *** [−14.1–−4.0]	3.1 [−2.2–8.5]	2.2 * [0.3–4.1]	2.4 [−0.5–5.2]	1.4 [−0.6–3.3]
**Smoking (ref. never)**					
Past smoking	−3.2 [−7.5–1.2]	1.7 [−2.8–6.3]	−0.4 [−1.8–1.0]	1.3 [−1.2–3.8]	0.5 [−1.2–2.2]
Smoking	−10.2 *** [−15.0–−5.5]	3.6 [−1.5–8.7]	0.6 [−1.2–2.3]	3.2 * [0.4–5.9]	2.9 ** [0.8–5.1]
**Binge drinking (once a month or more, ref. no)**					
Yes	−4.4 * [−8.6–−0.1]	5.1 * [0.6–9.6]	−0.8 [−2.1–0.5]	−0.5 [−2.8–1.9]	0.5 [−1.2–2.2]

^+^*p* < 0.1, * *p* < 0.05, ** *p* < 0.01, *** *p* < 0.00.

## Data Availability

The data used for the study are not publicly shared due to data protection laws. The data can be applied from the data holders, following data protection laws.

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
