# Peer review of "Pain in Multiple Sites and Clusters of Cause-Specific Work Disability Development among Midlife Municipal Employees"

_ijerph, 2021, doi:10.3390/ijerph18073375_

Round 1
Reviewer 1 Report
Overview: This is a great study to look at longitudinal effects of work disability broken down by type, sex, and other factors. The tables and graphics make it especially easy to understand. The authors show adequate research into the literature as well as thorough thought as to the limitations of the study.
Abstract: Good with adequate information regarding the study, population, and findings.
Introduction:
Admittedly, the introduction seems a bit short so I question whether a full literature was completed, but the findings are concise and thorough and certainly help to contextualize this study. I would recommend adding additional references or resources that have been found in other countries or areas to help contextualize the findings on a global scale in addition to regional.
Line 33 – make sure the time-to-first-long-term is appropriately spaced with hyphens
Materials and Methods:
Line 62 – 63 - although the cohort is described elsewhere, a brief overview of the cohort and sample would be beneficial for this audience.
Author Response
Overview: This is a great study to look at longitudinal effects of work disability broken down by type, sex, and other factors. The tables and graphics make it especially easy to understand. The authors show adequate research into the literature as well as thorough thought as to the limitations of the study.
Abstract: Good with adequate information regarding the study, population, and findings.
*************************
Response: Thank you for your assessment and these positive and constructive comments.
*************************
Introduction:
Admittedly, the introduction seems a bit short so I question whether a full literature was completed, but the findings are concise and thorough and certainly help to contextualize this study. I would recommend adding additional references or resources that have been found in other countries or areas to help contextualize the findings on a global scale in addition to regional.
*************************
Response: We agree with this point. As suggested, we have added more explanation regarding the previous literature. We have also added explanation how this study relates to our previous work. We now write:
“Previous studies, mainly analysing either the number of sickness absence (SA) or time to first long-term work disability event (DP), have shown that the locations, number of areas and intensity of pain contributes to the risk of subsequent long-term work disability [1-5]. A Finnish nationally representative study found that the number of musculoskeletal pain sites predict a higher risk of a disability pension award in a dose-response fashion [5]. Similar findings were reported in a large population-based Norwegian study which showed that widespread pain predicted a substantially higher risk of work disability [2]. However, the previous studies on pain and work disability have rarely analysed sickness absence and disability pensions simultaneously, taken the heterogeneity in the work disability development into account or analysed specific pain areas.”
*************************
Line 33 – make sure the time-to-first-long-term is appropriately spaced with hyphens
*************************
Response: Thank you so much for catching this issue. This is now corrected.
*************************
Materials and Methods:
Line 62 – 63 - although the cohort is described elsewhere, a brief overview of the cohort and sample would be beneficial for this audience.
*************************
Response: Thank you for pointing out this issue. We have added additional explanation regarding the cohort:
“The sample included for the analyses consisted of 2,878 initially full-time employees aged 45–57 in 2007, derived from the Helsinki Health Study (HHS). This cohort study is described in detail elsewhere [13]. In brief, HHS focusses on midlife employees of the City of Helsinki, the largest employer and municipality in Finland. The original baseline of the study was in 2000-2 when mailed questionnaires were sent to all employees of the City of Helsinki reaching their 40, 45, 50, 55 and 60 birthdays in each year. The cohort includes the employer’s and national register linkages for the respondents who provided consents for such linkages. This analysis uses a follow-up survey conducted in 2007 as the baseline for this study given that data on cause-specific sickness absence was not available immediately after the original baseline survey in 2000-2.”
*************************
Reviewer 2 Report
The manuscript entitled “Pain in multiple sites and clusters of cause-specific work disability development among midlife municipal employees” investigates to what extent pain in multiple sites and common risks factors related to work environment, occupational class and health behaviours are associated with cause-specific work disability (WD) development clusters in Finnish population cohort.
The manuscript is interesting, relevant to the disabilities topic and is well written and nicely designed with relatively good sample size.
However, the manuscript needs further assessment of next points:
- the pain that could be identified as a subjective feeling or is measured with any objective criteria, to specify objective criteria.
- location of the pain might be also mistakenly identified, especially in head pain.
- the age range of included people is in the midlife range. Have you considered menopause in those women, since hormone status might affect also the pain sensitivity? Additional factor missing: family status (number of children and related stress)
- Missing point is the strength of the sensitivity and not describe only locations (e.g. how many times per week or month, or continuous pain, especially with head located pain (migraine?).
Author Response
The manuscript entitled “Pain in multiple sites and clusters of cause-specific work disability development among midlife municipal employees” investigates to what extent pain in multiple sites and common risks factors related to work environment, occupational class and health behaviours are associated with cause-specific work disability (WD) development clusters in Finnish population cohort.
The manuscript is interesting, relevant to the disabilities topic and is well written and nicely designed with relatively good sample size.
*************************
Response: Thank you for your assessment and these positive comments.
*************************
However, the manuscript needs further assessment of next points:
- the pain that could be identified as a subjective feeling or is measured with any objective criteria, to specify objective criteria.
- location of the pain might be also mistakenly identified, especially in head pain.
- the age range of included people is in the midlife range. Have you considered menopause in those women, since hormone status might affect also the pain sensitivity? Additional factor missing: family status (number of children and related stress)
- Missing point is the strength of the sensitivity and not describe only locations (e.g. how many times per week or month, or continuous pain, especially with head located pain (migraine?).
*************************
Response: Thank you for these important points. We have added discussion about these limitations in the limitation sections of the paper.
“Our measure of pain is not without limitations. Pain is a subjective experience and it could be a symptom of a condition [6]. The experience of pain was asked in the surveys and was thus not assessed by a health professional. Furthermore, we did not have detailed data on the recurrence of pain and the length of the pain episodes. Lastly, our findings regarding pain and work disability should be considered in the context of midlife, mainly female public sector employees. However, multisite pain has shown similar associations with work disability in a cohort representative of all Finnish employees [5, 7].”
************************
Reviewer 3 Report
This paper is a professionally written and executed paper, although the results are not overly surprising.
The use of clusters to measure work disability warrants some more explanation.
Why cannot for instance an index or some other measurement be used?
What are the benefits of using these clusters? Please explain the use of clusters more accurately
Author Response
This paper is a professionally written and executed paper, although the results are not overly surprising.
*************************
Response: Thank you for these important points.
************************
The use of clusters to measure work disability warrants some more explanation.
Why cannot for instance an index or some other measurement be used?
What are the benefits of using these clusters? Please explain the use of clusters more accurately
*************************
Response: Thank you for these important points. We have extended our explanation regarding the cluster analysis:
“Sequence analysis techniques are often used, for example, to analyse employment patterns after a vocational rehabilitation [8-10]. However, the use of sequence analysis to investigate cause-specific work disability development is still limited. The heterogeneity in the diagnostic causes, interrelations, transitions, timing and the persistence of work disability is challenging to summarize while using traditional expected number of spells or time-to-event type modelling approaches. In contrast, sequence analysis can summarize this heterogeneity by creating meaningful groups of individuals with a similar cause-specific work disability development over time [11]. The method can take the duration of different states into consideration [11], which implies that a data driven distinction between temporary and permanent work disability can be made. Simplifying cause-specific work disability development to meaningful clusters, and identifying their determinants can provide added value for potential tailored prevention strategies.”
************************
Round 2
Reviewer 2 Report
The manuscript is nicely improved and now can be considered for the publication.